# Clinical manifestations, associated risk factors and treatment outcomes of Chronic Pulmonary Aspergillosis (CPA): Experiences from a tertiary care hospital in Lahore, Pakistan

**Waqas Akram** [1,2]*, **Muhammad Bilal Ejaz**[2,3], **Tauqeer Hussain Mallhi**[4], **Syed Azhar bin Syed Sulaiman**[1], **Amer Hayat Khan**[1]*

1 Discipline of Clinical Pharmacy, School of Pharmaceutical Sciences, Universiti Sains Malaysia, Penang, Malaysia, 2 Faculty of Pharmacy, University of Central Punjab, Lahore, Punjab, Pakistan, 3 Gulab Devi Teaching Hospital, Lahore, Pakistan, 4 Department of Clinical Pharmacy, College of Pharmacy, Jouf University, Al-Jouf, Kingdom of Saudi Arabia

* waqas.akram@student.usm.my (WA); dramer@usm.my (AHK)

## Abstract

### Background

Chronic pulmonary aspergillosis (CPA) has a wide spectrum of illnesses depending on the progression of the disease and comorbid conditions. However, there is an inadequacy of investigations regarding clinical, laboratory, risk factor and prognostic data on CPA. The current study is aimed to consider the clinical manifestations, risk factors and outcomes of CPA.

### Methodology

Retrospective records of all patients with a confirmed diagnosis of CPA who sought treatment at Gulab Devi Chest Hospital Lahore, Pakistan from January 2017 to December 2019 were evaluated. Data regarding demographics, clinical manifestations, comorbidities, radiographic and microbiological findings, length of hospital stay (LOS) and intensive care unit (ICU) admission was collected and analyzed to identify the factors associated with mortality. The independent factors associated with mortality were also identified by appropriate analyses.

### Results

A total of 218 CPA patients were included in this study. The mean age was 45.75 ± 6.26 years. Of these, 160 (73.4%) were male, and 65 (29.8%) had diabetes. The mean LOS was 18.5 ± 10.9 days. The most common type of CPA was simple aspergilloma (56%) followed by chronic cavitary pulmonary aspergillosis (CCPA) (31.2%). About one half of the patients had a history of pulmonary tuberculosis (TB) and treatment response rates were low in patients with active TB. The overall mortality rate was 27.1%. ICU admission was required

**Data Availability Statement:** All relevant data are within the manuscript and its Supporting Information files.

**Funding:** The author(s) received no specific funding for this work.

**Competing interests:** The authors have declared that no competing interests exist.

for 78 (35.8%) patients. Diabetes mellitus (DM), hematological malignancies and chronic kidney disease (CKD) were the common underlying conditions predicting a poor outcome. Mean LOS, hematological malignancies, consolidation and ICU admission were identified as the independent factors leading to mortality.

## Conclusions

CPA had a significant association with TB in the majority of cases. Treatment response rates in cases with active TB were comparatively low. Cases with high mean LOS, hematological malignancies, consolidation, ICU admission, CKD and DM experienced poor outcomes. High mean LOS, hematological malignancies, consolidation and ICU stay were identified as independent risk factors for mortality. Future large prospective studies, involving aspergillus specific immunoglobulin G (IgG) antibody testing, are required for a better understanding of CPA in Pakistan.

## Introduction

Aspergillus species are ubiquitous in the environment and exposure to the conidia is common. However, only a minority of people develop clinical disease, and the development of disease is often determined by host characteristics, e.g., immune status, genetic predisposition, underlying lung pathology and prior pulmonary infection such as tuberculosis (TB) [1].

Classically, chronic pulmonary aspergillosis (CPA) in immunocompetent patients presents as a saprophytic infection in a pre-existing cavity, often following an infection such as TB or prior lung surgery [2]. The key features of CPA include steady destruction of lung tissue showing progressive cavity formation, fibrosis, and pleural thickening. The syndrome is mostly seen in patients with lung cavitation diseases or structural pulmonary abnormalities [3].

There are a number of recognized manifestations of CPA: subacute invasive pulmonary Aspergillosis (SAIA) (which may be referred to as chronic necrotizing pulmonary aspergillosis (CNPA)), chronic cavitary pulmonary aspergillosis (CCPA), and chronic fibrosing pulmonary aspergillosis (CFPA) [4]. SAIA occurs in the cases with some degree of immune compromise, and may present with nodules, consolidation or cavitation upon chest imaging, and a more rapidly progressing clinical course. CCPA presents with single or multiple cavities, with or without aspergilloma(s), and CFPA has the same appearance with the added features of pulmonary fibrosis, which may be progressive and destructive [2].

Estimates of the incidence and prevalence of CPA are difficult; however, the global burden of disease is increasing [5, 6]. The prevalence of CPA has been estimated at 3,000,000 cases worldwide [7]. In 2017, a review encompassing data from 43 countries revealed the highest incidence was estimated in Russia (126.9 cases/100,000) followed by Philippines, Nigeria (78 cases/100,000), Pakistan (70 cases/100,000) and Vietnam (61 cases/100,000). The overall incidence for all the countries included in the study was 22 cases/100,000 (14.2–30.59, 95% CI) [8]. In Pakistan, the risk of development of CPA is high because CPA is regarded as the most severe sequelae of Pulmonary Tuberculosis (PTB) [5], and Pakistan ranks 5th among the countries with the highest PTB burden [9]. The most recent burden estimate of CPA in Pakistan was 39 cases/100,000 people [10].

This data makes CPA as a disease of significant concern for a high pulmonary TB burden countries like Pakistan. The situation is made more complicated by the poor fungal diagnostic

capabilities in most of the laboratories in Pakistan, the emergence of antifungal resistance, a lack of antimicrobial stewardship, poor infection control practices and the lack of availability of essential antifungal agents [11]. There is a paucity of data on CPA and very few studies are available providing data regarding clinical manifestations, underlying conditions and risk factors [10, 12]. Therefore, a retrospective study was designed at a tertiary care hospital in Lahore, Pakistan to highlight the clinical manifestations, underlying conditions, predictors associated with mortality and risk factors of CPA.

## Methods

### Study design and settings

This was a retrospective study conducted at Gulab Devi Chest Hospital, which is a tertiary care facility in Lahore, Pakistan. All of the following patients who enrolled for treatment at the facility from January 1st 2017 to December 30th 2019 were included in this study: (1) those aged 18 years or above; (2) with a CPA diagnosis, including all of its types, i.e., simple aspergilloma, CCPA, SAIA and CFPA; (3) with positive bronchoalveolar lavage (BAL) and/or sputum cultures for *Aspergillus* Spp. and/or positive pulmonary histopathology suggesting the presence of *Aspergillus* Spp; and (4) with a chest CT scan/ X-Ray suggestive of aspergillosis.

Patients with other forms of pulmonary aspergillosis than CPA, and those with incomplete records, were excluded from the study. Patients with positive cultures but suggesting colonization only were also excluded because the colonization by microorganisms does not cause disease by itself [13].

### Ethical approval

The study was approved by the Institutional Review Board (IRB) of Gulab Devi Chest Hospital, Lahore. A written informed consent regarding the use of their clinical records was obtained from patients visiting the study site for their treatment. The hospital's IRB waived the requirement for informed consent from all those patients who were critically ill or deceased.

### Data collection

A validated data collection form was used to collect patients' demographics, associated conditions, underlying lung pathologies, radiological and microbiological findings, length of hospital stay and requirement for ICU admission. The hospital's computerized record system and patients' files were evaluated for the purpose of data collection.

### Operational definitions

Patients were categorized further on the basis of the types of CPA, namely: CCPA, SAIA, CFPA and aspergilloma. CPA is defined as the presence of the following for at least three months, along with chronic respiratory symptoms: (1) one or more cavities either with or without a fungal ball or nodules/pleural thickening seen on radiographic imaging; (2) direct evidence of infection by *Aspergillus* Spp. revealed by either microscopy, a culture taken from biopsy, or both; and (3) the presence of an immunological response to *Aspergillus* antigens and the exclusion of an alternative diagnosis [14]. Simple pulmonary aspergilloma is defined as the presence of a single cavity with a single fungal ball and no spread or progression seen in radiological imaging over the course of at least 3 months of observation. CCPA is defined as the presence of one or multiple cavities in the lungs, possessing one or multiple fungal balls, along with microbiological evidence suggesting the presence of *Aspergillus* Spp., and sufficient symptomatic and/or radiographic evidence of an increase in number of cavities and/or infiltrates

over at least 3 months of observation [12]. SAIA is defined as the radiographic findings suggestive of the presence of hyphae penetrating into the lung tissue [14]. The label CFPA is assigned to cases with one or multiple cavities, either with or without aspergilloma, that have an additional feature of pulmonary fibrosis which may progressively destroy the lung tissue. According to the Fleischner Society: Glossary of Terms for Thoracic Imaging, a nodule is defined as a rounded, opaque, well or poorly defined mass, with a diameter of up to 3 centimeters [2].

Outcomes were defined as the observable or measurable changes in health status, which included ICU admission and mortality. A deterioration of health status was identified among living patients developing respiratory complications and staying in the ICU till the end of the study. Recovered or stable patients were those patients either not requiring ICU admission or those that may have developed respiratory complications and were admitted to the ICU but recovered and were discharged from the ICU before the completion of the study. Both mortality and deterioration were considered to be poor outcomes, while stability of condition or recovery were considered as favorable outcomes.

## Diagnosis of CPA

Fungal cultures were carried out by well-trained laboratory technologists and were reviewed by consultant microbiologists in a suitable and well-controlled environment. The culture plates were examined properly for 4 weeks prior to being reported as negative. *Aspergillus* Spp. was identified on the basis of physical appearance: the microscopic and morphological characteristics of the colonies [15]. The diagnosis made was in accordance with the ICD (International Disease Classification) that is the most extensively used nosology [16]. ICD code B44, that describes the diagnosis of Aspergillosis, was observed.

## Statistical methods

SPSS (Release: 23.0, Standard version, Copyright SPSS; 1989–2015) was used to carry out all the statistical analyses. Quantitative variables (age, length of hospital stay) and qualitative variables (such as gender, diet conditions, smoking status, comorbidities, symptoms, risk factors, respiratory complications, microbiological findings, radiographic findings, and requirement for admission into the ICU) were analyzed by descriptive statistics. Comparisons between outcomes (that were Stable/Recovered, Deteriorated and Mortality) as dependent variables and other independent variables were made using the Chi-square test. The parameters for univariate analysis were selected based on their biological and clinical plausibility, and as indicated by the previous literature. The test variables reaching significance on univariate analysis were analyzed via multinomial logistic regression analysis to determine the independent factors predicting the mortality. From the parameter estimates, p-value and odds ratio along with 95% confidence interval (CI) were taken into consideration. All the p-values $\leq 0.05$ were considered significant.

## Results

### Characteristics of patients

A total of 521 cases with confirmed diagnosis of aspergillosis were reviewed. Of these, 218 (41.84%) cases met the inclusion criteria. The mean age of participants was 45.75 ± 6.26 years and 160 (73.4%) were males (**Table 1**). Of the total 218 CPA cases, 122 (56%) were diagnosed with simple aspergilloma, 68 (31.2%) were diagnosed with CCPA and 28 (12.8%) had SAIA. There were no cases of aspergillus nodules and CFPA in our study population (Fig 1).

**Table 1. Characteristics of patients with chronic pulmonary aspergillosis.**

| Characteristics of patients. | Out of 218 (%) |
|---|---|
| Mean age ± SD (years) | 45.75 ± 6.26 |
| **Gender:** | |
| Male | 160 (73.4%) |
| Female | 58 (26.6%) |
| **Smoking status:** | |
| Non-smokers | 106 (48.6%) |
| Ex-smokers | 64 (29.4%) |
| Current smokers | 48 (22%) |
| **Symptoms:** | |
| Cough | 207 (95%) |
| Fatigue | 202 (92.7%) |
| Sputum production | 197 (90.4%) |
| Fever | 187 (85.8%) |
| Hemoptysis | 130 (59.6%) |
| Weight loss | 76 (34.9%) |
| Dyspnea | 39 (17.9%) |
| Chest pain | 22 (10.1%) |
| **Associated Conditions:** | |
| Diabetes mellitus | 65 (29.8%) |
| Chronic kidney disease | 61 (28%) |
| Hematological malignancies | 36 (16.5%) |
| Chronic liver disease | 12 (5.5%) |
| Neutropenia | 19 (8.7%) |
| Steroid use | 147 (67.4%) |
| Inhaled | 32/147 (21.8%) |
| Oral | 115/147 (78.2%) |
| Chemotherapy | 16 (7.3%) |
| **Pulmonary Tuberculosis:** | |
| Previous TB | 96 (44%) |
| Active TB | 41 (18.8%) |
| **Other Respiratory Conditions:** | |
| Pulmonary sarcoidosis | 46 (21.1%) |
| Bronchiectasis | 33 (15.1) |
| Asthma | 23 (10.6%) |
| COPD | 23 (10.6%) |
| ILD | 10 (4.6%) |
| **Types of CPA Encountered:** | |
| Simple Aspergilloma. | 122 (56%) |
| CCPA | 68 (31.2%) |
| SAIA | 28 (12.8%) |

TB: Tuberculosis, COPD: Chronic obstructive pulmonary disorder.

ILD: Interstitial Lung Disease, CPA: Chronic Pulmonary Aspergillosis. CCPA: Chronic Cavitary Pulmonary Aspergillosis, SAIA: Subacute Invasive Aspergillosis. SD: Standard Deviation.

Several non-specific clinical manifestations were observed in the current study with a cough being the most common [n = 207 (95%)], followed by fatigue [n = 202 (92.7%)].

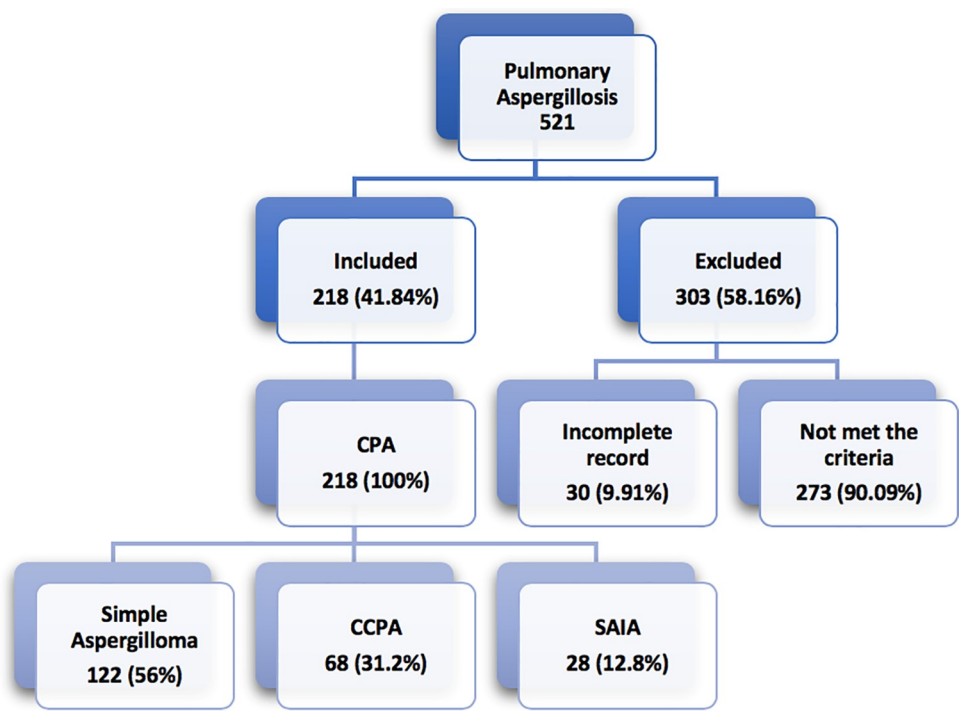

**Fig 1. Enrollment of patients with pulmonary aspergillosis.**

## Associated lung diseases

PTB was found to be the most common associated lung disease (with n = 137, 62.8%). Out of 137 cases, 96 (70%) previously had tuberculosis and 41 (29.9%) had active TB disease (**Table 1**). Out of the 122 cases with simple aspergilloma, 20 (16.4%) had active TB, out of which 2 (10%) had underlying chronic obstructive pulmonary disorder (COPD). In 68 cases of CCPA and 28 cases of SAIA, the presence of active TB was 14 (20.6%) and 7 (25%), respectively, of which 1 of 14 (7.1%) and 1 of 7 (14.3%) cases respectively had COPD. Furthermore, pulmonary sarcoidosis was seen in 46 (21.1%) cases.

## Underlying conditions

Diabetes mellitus was reported as the most common underlying condition with n = 65 (29.8%) followed by chronic kidney disease (n = 61, 28%). Hematological malignancies and chronic liver disease were present in 36 (16.5%) and 12 (5.5%) cases, respectively (**Table 1**). Out of 68 patients with CCPA, 23 (33.8%) had diabetes. Out of 28 patients with SAIA and 122 patients with simple aspergilloma, 8 (28.6%) and 34 (27.9%) had diabetes, respectively.

   Steroid use (n = 147; 67.4%) was a common associated condition. Out of 147 cases with steroid use, 115 (78.2%) received oral prednisolone, with a mean dose of 35.1 ± 8.3 milligrams for 4.5 ± 1.1 weeks, followed by a tapered dose. Inhaled corticosteroids were used in 32 out of 147 (21.7%) patients, with a mean duration of 3.5 ± 0.6 months, out of which 13 (40.6%) received fluticasone at the mean initial dose of 134.6 ± 47.3 micrograms with the mean maximum dose of 546.1 ± 210.6 micrograms, and 19 (59.4%) received beclomethasone at the mean initial dose of 66.8 ± 22.1 micrograms with a mean maximum dose of 480 ± 127.7 micrograms, and with the dose tapered thereafter.

## Laboratory, microbiological and radiological findings

Microbiological findings revealed that BAL/sputum cultures were positive for *Aspergillus* Spp. in 185 (84.9%) cases while 111 (50.9%) cases had positive histopathology. A total of 11 (5.04%) cases had positive histopathology only, while 85 (38.99%) cases had positive BAL/sputum cultures only. Both histopathology and cultures were positive in 100 (45.9%) cases. *Aspergillus fumigatus* was the most common isolate (n = 100, 45.9%) (Table 2).

Inflammatory markers, i.e., Erythrocyte sedimentation rate (ESR) and C-Reactive protein (CRP), were also elevated, with a median of 13.7 mg/L (IQR: 9.3–17.5) and 67 mm/h (IQR: 55–78), respectively. Upon chest imaging, 143 (65.6%) showed fungal balls, and nodular

**Table 2. Laboratory, microbiological and radiological findings.**

| Characteristics | Out of 218 (%) |
|---|---|
| **Laboratory Findings:** | |
| C-Reactive Protein [Median (IQR)] (mg/L) | 13.7 (9.3–17.5) |
| CRP >3–9 | 26 (11.9%) |
| CRP >9–18 | 149 (68.3%) |
| CRP >18 | 43 (19.7%) |
| Erythrocyte Sedimentation Rate [Median (IQR)] (mm/h) | 67 (55–78) |
| ESR > 20–50 | 21 (9.6%) |
| ESR > 50–80 | 147 (67.4%) |
| ESR > 80–100 | 45 (20.6%) |
| ESR >100 | 5 (2.3%) |
| **Positive Cultures:** | |
| Sputum/BAL | 185 (84.9%) |
| Histopathology | 111 (50.9%) |
| Sputum/BAL only | 85 (38.99%) |
| Histopathology only | 11 (5.04%) |
| Both sputum/BAL and Histopathology | 100 (45.9%) |
| Isolated *Aspergillus* spp. | |
| *Aspergillus fumigatus* | 100 (45.9%) |
| *Aspergillus flavus* | 81 (37.2%) |
| *Aspergillus niger* | 20 (9.2%) |
| *Aspergillus terreus* | 17 (7.8%) |
| **Radiological Findings:** | |
| Fungal ball | 143 (65.6%) |
| Nodular infiltrates | 93 (42.7%) |
| Consolidation | 73 (33.5%) |
| Pleural thickening | 53 (24.3%) |
| Bronchiectasis | 33 (15.1%) |
| Number of cavities: | |
| 1 | 123 (56.4%) |
| ≥2 | 95 (43.6%) |
| Location of cavities: | |
| Bilateral | 95 (43.6%) |
| Left lobe | 65 (29.8%) |
| Right lobe | 58 (26.6%) |

BAL: Bronchoalveolar Lavage, CRP: C-Reactive Protein, ESR: Erythrocyte Sedimentation Rate. IQR: Interquartile Range.

infiltrates were seen in 93 (42.7%) cases. Consolidation was present in 73 (33.5%) cases. Pleural thickening was observed in 53 (24.3%) cases and bronchiectasis was seen in 33 (15.1%) cases. A single cavity was present in 123 (56.4%) cases, and 95 (43.6%) cases had two or multiple cavities. In 95 (43.6%) cases, the cavities were bilaterally located. Cavities in the left and right lobes were observed in 65 (29.8%) and 58 (26.6%) cases, respectively (**Table 2**).

## Antifungal therapy and outcomes

Out of 218 cases, itraconazole was used in 152 (69.7%) cases and 51 (23.4%) were treated with amphotericin B. In the remaining 15 (6.9%) cases, voriconazole was used as an anti-fungal agent. The median duration of treatment was 6.5 (IQR: 6.0–8.0) months. The duration of treatment in 114 (52.3%) cases was more than 6 months, while 95 (43.6%) received treatment for at least 6 months. The median duration of follow up remained 7.5 (IQR: 6.0–8.5) months. Improvement in symptoms was observed in 122 (52.3%) cases. Radiographic changes such as pleural thickening and nodular infiltrates were observed in 41 (18.8%) and 71 (33%) cases, respectively, as compared to the presence of pleural thickening and nodular infiltrates in 53 (24.3%) and 93 (42.7%) cases, respectively, before the initiation of therapy. A decline in inflammatory markers, i.e., ESR and CRP, was seen in 188 (86.2%) and 177 (81.2%) cases, respectively (**Table 3**).

Respiratory complications were developed in 129 (59.2%) cases. Hypoxic respiratory failure was the most common respiratory complication, followed by pneumothorax. ICU admission was required in 78 (35.8%) cases. Stability or recovery was observed in 119 (54.6%) patients, whereas the clinical conditions of 40 (18.3%) patients were deteriorated. Mortality was seen in 59 (27.1%) patients. Out of the CPA variants, the highest mortality was seen in SAIA [n = 9/28, (32.1%)] (**Table 3**).

## Adverse effects associated with anti-fungal therapy

Adverse effects related to antifungal therapy were observed, which led to the discontinuation of treatment in 8 (3.7%) cases after a median of 2.5 months (IQR: 2.1–3.05). The majority of adverse effects were related to GIT, such as nausea (n = 51; 23.4%), diarrhea (n = 26; 11.9%) and anorexia (n = 60; 27.5%). Hepatotoxicity (liver transaminase level of $\geq$ 120 IU/L) was observed in 17 (7.8%) cases (**Table 3**).

## Impact of prognosis in patients with TB

A proportion of our study population (n = 41; 18.8%) had active tuberculosis disease. The mean age of patients with the collective presence of TB and CPA was 45.9 years. Amphotericin B was the antifungal agent of choice for CPA in all the 41 cases with active tuberculosis disease. All the cases with active TB disease received standard anti-TB therapy, i.e., a combination of Ethambutol HCL, Rifampicin, Isoniazid and Pyrazinamide. The treatment response rate, based on the alleviation of symptoms, in the patients with co-existing tuberculosis was 51.2%, as compared to 56.3% and 58% in the patients having tuberculosis previously and in those with no tuberculosis, respectively. Out of 41 patients, respiratory complications were developed in 23 (56%) cases. ICU admission was required in 15 (36.6%) cases and mortality was seen in 11 (26.8%) patients (**Table 4**).

## Factors associated with mortality

Underlying conditions such as hematological malignancies ($p$ = 0.03), chronic kidney disease ($p$ = 0.03), and diabetes mellitus ($p$ = 0.04) were significant factors associated with mortality.

**Table 3. Responses to therapy and outcomes.**

| Characteristics | Out of 218 (%) |
|---|---|
| **Antifungal agent used** | |
| Itraconazole | 152 (69.7%) |
| Amphotericin B | 51 (23.4%) |
| Voriconazole | 15 (6.9%) |
| **Duration of Treatment [Median (IQR)] (Months)** | 6.5 (6.0–8.0) |
| <6 Months | 9 (4.1%) |
| At least 6 months | 95 (43.6%) |
| >6 Months | 114 (52.3%) |
| **Mean LOS ± SD (Days)** | 18.5 ± 10.9 |
| <10 | 16 (7.3%) |
| 10–19 | 124 (56.9) |
| 20–29 | 20 (9.2%) |
| 30–40 | 46 (21.1%) |
| >40 | 12 (5.5%) |
| **Changes after treatment** | |
| Improvement in symptoms | 122 (56%) |
| Pleural thickening | 41 (18.8%) |
| Nodular infiltrates | 72 (33%) |
| Decrease in CRP | 177 (81.2%) |
| Decrease in ESR | 188 (86.2%) |
| Negative conversion of sputum/BAL cultures | 67/185 (36.3%) |
| **Respiratory complications** | 129 (59.2%) |
| Hypoxic respiratory failure | 59 (27.1%) |
| Pneumothorax | 51 (23.4%) |
| Bronchopleural fistula | 4 (1.8%) |
| Empyema | 4 (1.8%) |
| **Outcomes** | |
| Deteriorated | 40 (18.3%) |
| Stable/Recovered | 119 (54.6%) |
| Overall mortality | 59 (27.1%) |
| Mortality in SA | 29 out of 122 (23.8%) |
| Mortality in CCPA | 21 out of 68 (30.9%) |
| Mortality in SAIA | 9 out of 28 (32.1%) |
| Adjunctive surgical resection | 19 (8.7%) |
| ICU admission | 78 (35.8%) |
| Follow up duration [Median (IQR)] (Months) | 7.5 (6.0–8.5) |
| **Adverse effects due to antifungal therapy** | |
| Nausea | 51 (23.4%) |
| Diarrhea | 26 (11.9%) |
| Anorexia | 60 (27.5%) |
| Hepatotoxicity | 17 (7.8%) |

LOS: Length of stay at hospital, SD: Standard deviation, IQR: Interquartile Range

ICU: intensive care unit, SA: Simple aspergilloma, CCPA: Chronic cavitary pulmonary aspergillosis, SAIA: Subacute invasive aspergillosis CRP: C-Reactive Protein

ESR: Erythrocyte Sedimentation Rate.

**Table 4. Impact on prognosis of patients with TB.**

| Characteristics | Out of 41 (%) |
|---|---|
| Mean age ± SD (years) | 45.90 ± 6.34 |
| Age distribution | |
| <35 | 3 (7.3%) |
| 35–45 | 17 (41.5%) |
| >45 | 21 (51.2%) |
| Treatment Choice: | |
| Amphotericin B | 41 (100%) |
| Improvement in symptoms | 21 (51.2%) |
| Stable/Recovered | 20 (48.8%) |
| Deteriorated | 10 (24.4%) |
| Mortality | 11 (26.8%) |
| Adjunctive surgical resection | 6 (14.6%) |
| Respiratory complications | 23 (56.1%) |
| Hypoxic respiratory failure | 12 (29.3%) |
| ICU admission | 15 (36.6%) |

SD: Standard Deviation, ICU: Intensive Care Unit.

Admission to ICU ($p$ = <0.001) was also found to be significantly associated with mortality, as 29 (49.2%) out of 59 cases ending in mortality required ICU admission, and only 10 (12.8%) out of 78 patients requiring ICU admission recovered. Hypoxic respiratory failure ($p$ = 0.028) was the most significant factor leading to ICU admission.

Furthermore, higher LOS ($p$ = 0.007), consolidation ($p$ = 0.02) and development of respiratory complications ($p$ = 0.03) were also significant factors associated with mortality. In light of the analysis performed, recovery was seen in 11 out of 16 patients for which the length of hospital stay was < 10 days. Among 124 patients with LOS in the range of 10–19 days, recovery was seen in 81 (65.3%) cases. Whereas the recovery rates significantly declined once the length of stay in hospital was over 20 days. Age distribution ($p$ = 0.012), chronic liver disease ($p$ = 0.003), consolidation ($p$ = 0.016) and diabetes mellitus ($p$ = 0.05) were the significant factors leading to prolonged LOS. Among respiratory complications, hypoxic respiratory failure was the most significant factor associated with mortality ($p$ = 0.04). Independent factors associated with mortality on logistic regression analysis were mean LOS, hematological malignancies, consolidation and ICU admission (**Table 5**).

## Discussion

This study provided valuable information regarding the clinical manifestations, underlying conditions, associated risk factors and outcomes of CPA. The literature suggests that out of all the forms of CPA, CCPA is the most common [14]. However, in this study, the most encountered form was simple aspergilloma. The reason for this is the site of the study which was a tertiary care hospital, because simple aspergilloma is a late and progressed feature of CPA and the majority of cases are referred for further interventions in a tertiary care hospital [17].

Microbiological findings revealed that the percentage of positive sputum/BAL cultures was 84.9%, which was higher than the previous study carried out at a tertiary care hospital in Karachi, Pakistan, that reported 66.7% positive sputum/BAL cultures [12]. The high percentage may also be attributed to colonization or in-process contamination. In an attempt to reduce and possibly eliminate the risk of colonization affecting the culture results to some extent, the

**Table 5. Factors associated with outcomes and independent risk factors associated with Mortality.**

| Factors | Outcomes | | | p-value* |
|---|---|---|---|---|
| | Stable/Recovered (n = 119) | Deteriorated (n = 40) | Mortality (n = 59) | |
| Gender | | | | 0.559 |
| Male | 84 (70.6%) | 30 (75%) | 46 (78%) | |
| Female | 35 (29.4%) | 10 (25%) | 13 (22%) | |
| Age Distribution (years) | | | | 0.338 |
| <35 | 6 (5%) | 5 (12.5%) | 3 (5.1%) | |
| 35–45 | 53 (44.5%) | 14 (35%) | 21 (35.6%) | |
| >45 | 60 (50.4%) | 21 (52.5%) | 35 (59.3%) | |
| Diet conditions | | | | 0.396 |
| Well-nourished | 20 (16.8%) | 8 (20%) | 15 (25.4%) | |
| Malnourished | 99 (83.2%) | 32 (80%) | 44 (74.6%) | |
| Smoking status | | | | 0.083 |
| Current smoker | 22 (18.5%) | 6 (15%) | 20 (33.9%) | |
| Ex-smoker | 33 (27.7%) | 15 (37.5%) | 16 (27.1%) | |
| Non-smoker | 64 (53.8%) | 19 (47.5%) | 23 (39%) | |
| Mean LOS (± SD) (days) | 15.68 (± 9.19) | 14.92 (± 08) | 26.64 (± 11.7) | 0.007 |
| <10 | 11 (9.2%) | 5 (12.5%) | 0 (0.0%) | |
| 10–19 | 81 (68.1%) | 26 (65%) | 17 (28.8%) | |
| 20–29 | 8 (6.7%) | 3 (7.5%) | 9 (15.3%) | |
| 30–40 | 15 (12.6%) | 6 (15%) | 25 (42.4%) | |
| >40 | 4 (3.4%) | 0 (0.0%) | 8 (13.6%) | |
| Treatment choice | | | | 0.584 |
| Itraconazole | 87 (73.1%) | 25 (62.5%) | 40 (67.8%) | |
| Voriconazole | 9 (7.6%) | 3 (7.5%) | 3 (5.1%) | |
| Amphotericin B | 23 (19.3%) | 12 (30%) | 16 (27.1%) | |
| Respiratory complications | 63 (52.9%) | 23 (57.5%) | 43 (72.9%) | 0.038 |
| Hypoxic respiratory failure | 25 (21%) | 11 (27.5%) | 23 (39%) | 0.040 |
| Pneumothorax | 27 (22.7%) | 7 (17.5%) | 17 (28.8%) | 0.412 |
| Bronchopleural Fistula | 4 (3.4%) | 0 (0.0%) | 0 (0.0%) | 0.184 |
| Empyema | 2 (1.7%) | 2 (5%) | 0 (0.0%) | 0.188 |
| Bronchiectasis | 17 (14.3%) | 4 (10%) | 12 (20.3%) | 0.344 |
| Consolidation | 33 (27.7%) | 12 (30%) | 28 (47.5%) | 0.028 |
| Pleural thickening | 26 (21.8%) | 8 (20%) | 19 (32.2%) | 0.247 |
| Fungal balls | 83 (69.7%) | 26 (65%) | 34 (57.6%) | 0.276 |
| ICU Admission | 10 (8.4%) | 39 (97.5%) | 29 (49.2%) | <0.001 |
| **Underlying conditions associated with mortality:** | | | | |
| Diabetes mellitus | 31 (26.1%) | 9 (22.5%) | 25 (42.4%) | 0.043 |
| Hematological malignancies | 16 (13.4%) | 4 (10%) | 16 (27.1%) | 0.032 |
| CKD | 27 (22.7%) | 10 (25%) | 24 (40.7%) | 0.038 |
| CLD | 7 (5.9%) | 2 (5%) | 3 (5.1%) | 0.965 |
| Neutropenia | 12 (10.1%) | 3 (7.5%) | 4 (6.8%) | 0.729 |
| Chemotherapy | 12 (10.1%) | 2 (5%) | 2 (3.4%) | 0.224 |
| Steroid use | 77 (64.7%) | 30 (75.0%) | 40 (67.8%) | 0.484 |
| Tuberculosis | | | | 0.778 |
| Active TB | 20 (16.8%) | 10 (25%) | 11 (18.6%) | |
| Previous TB | 52 (43.7%) | 16 (40%) | 28 (47.5%) | |
| COPD | 12 (10.1%) | 2 (5%) | 9 (15.3%) | 0.257 |

(*Continued*)

**Table 5.** (Continued)

| Factors | Outcomes | | | p-value* |
|---|---|---|---|---|
| | Stable/Recovered (n = 119) | Deteriorated (n = 40) | Mortality (n = 59) | |
| Asthma | 12 (10.1%) | 7 (17.5%) | 4 (6.8%) | 0.227 |
| Independent factors associated with mortality on logistic regression analysis: | | | | |
| Factors | OR (95% CI) | | | p-value** |
| Mean LOS (days) | 1.107 (1.063 to 1.154) | | | <0.01 |
| ICU admission | 12.806 (4.614 to 35.542) | | | <0.01 |
| Hematological malignancies | 3.747 (1.336 to 10.510) | | | 0.012 |
| Consolidation | 2.753 (1.184 to 6.400) | | | 0.019 |

*Chi-square.

**Multinomial Logistic Regression.

Bold: Significant, LOS: Length of hospital stay, CKD: Chronic Kidney Disease

CLD: Chronic liver disease, TB: Tuberculosis, COPD: Chronic obstructive pulmonary disease, ICU: Intensive care unit, SD: Standard Deviation.

results were concluded after duly accessing the significance of growth. The risk of contamination was significantly reduced by carrying out microbiological processes under a controlled environment and using safety cabinets. Furthermore, the most common isolate was *Aspergillus fumigatus*, which was the same as found by previous research [18–21].

According to the literature, risk factors for CPA include a history of mycobacterial illness, COPD, asthma and interstitial pneumonia [12, 22–24]. In our study, a history of steroid use was also found as a common associated condition, and association between corticosteroid use and development of pulmonary aspergillosis is also reflected by the previous studies [25, 26]. This study revealed that the majority of patients developed CPA as a post-TB sequel, which is also supported by a number of previous studies [5, 18, 22, 27, 28]. But a rare condition, that is the collective presence of active PTB and CPA, was also revealed, which was previously reported in very few studies [29–31]. DM was found to be the most significant comorbidity, as 29.8% cases were diabetic. CCPA had the highest percentage of patients with diabetes (33.8%) compared to other forms of CPA. The previous studies have also reported the association between DM and pulmonary aspergillosis infections, and individuals with DM are more prone to getting pulmonary aspergillus infections, even when the other risk factors are not present [32]. The increase in occurrence of tuberculosis is attributed to DM, possibly due to the deteriorated immune status of diabetic patients, and thus the clinical manifestations are worsened and treatment outcomes are adversely affected due to the collective presence of DM and TB [33–35]. In our study, 21.1% of patients had a history of pulmonary sarcoidosis, and it has also been documented that CPA can occur, complicating the pulmonary sarcoidosis [36]. The symptoms observed in this study were non-specific such as cough, fever, fatigue, weight loss and hemoptysis, as indicated in previous studies, and in a few cases, CPA has also been found to be asymptomatic [37, 38].

Radiographic findings revealed that the majority of patients had fungal balls. Nodular infiltrates, consolidation, bronchiectasis and pleural thickening were also observed. The presence of fungal balls was reported in 90% of CPA patients in a recent study [39]. A total of 56.4% CPA patients in our study had a single cavity as compared to 86% reported by a previous study, and most of the cavities in our study were bilaterally present unlike in the previous study [40]. The number of cases with nodular infiltrates and pleural thickening declined as a response to treatment.

The most common respiratory complication was hypoxic respiratory failure because CPA is a progressive lung disease. In CCPA, the compromised respiratory system is already well-

documented [41]. Mortality rates vary significantly between the studies because mortality is linked to the severity of the disease, various complications and delayed diagnosis. In a previous study, a mortality rate of 8% was reported with a median follow-up time of 10 months [42]. A comparatively recent study revealed a high mortality rate of 50% with a median follow-up time of 27.8 months [43]. This study revealed an overall mortality rate of 27.1% with a median follow-up time of 7.5 months.

In a previous study, based on the alleviation of symptoms, the overall treatment response rates involving the use of voriconazole in about 99% of the study population was 73% [40], as compared to 56% in our study. Antifungal therapy remained adequately acceptable for most of the patients as treatment discontinuation was seen in only 3.7% of cases as compared to 10% in a recent study [40]. Our study population had a considerable number of patients with co-existing tuberculosis; thus, Amphotericin B was considered as an anti-fungal agent of choice due to major interactions among triazoles and first line anti-tubercular drugs [14]. In the patients with co-existing pulmonary tuberculosis, the treatment response rate based on alleviation of symptoms was 51.2%, which was low compared to the treatment response rate in patients having TB previously and patients having no TB.

Mean LOS, admission to ICU, presence of hematological malignancies and consolidation were independent factors associated with the mortality, whereas in a previous study hypoxic respiratory failure, DM and mean LOS were the risk factors associated with mortality [12]. The presence of invasive fungal infections in patients with hematological malignancies is already well documented as the evidence of fungal infections at autopsy has already been found in 20–50% of the patients with such malignancies [44, 45]. This is possibly due to the enhanced risk of infection in patients with hematological malignancies leading to morbidity and mortality [46]. In our study, the non-survivors appeared to have a longer stay at hospital and required ICU admission, which is also evident in a recent study [12]. Due to the scarcity of data regarding mean LOS and ICU stay in patients with chronic pulmonary aspergillosis, a clear comparison is very difficult to make as most of the available data was related to the IPA.

## Study limitations

Though plenty of useful clinical information was extracted from this study, it had a few limitations, which should be considered while interpreting the results. Findings produced by a study conducted in a single center cannot be generalized, and the nature of the study was retrospective, so standardization was not ensured [2]. The latest diagnostic methods, particularly antigen detection techniques, were not available. Nowadays, a variety of diagnostic methods are available to detect fungal infections of the lungs. These methods include the antigen detection techniques, polymerase chain reactions (PCR) and various serological methods [47]. Neither the galactomannan (GM) tests from bronchoalveolar lavage fluid (BALF) nor the serum GM antigen tests have shown significant efficacy in the case of CPA [48, 49]. A research study concluded that the combination of GM and β-D-Glucan (βDG) tests performed on BALFs of those with a suspected infection was a more useful approach for the diagnosis of CPA compared to any test used alone [50–52]. Due to the unavailability of antigen detection techniques, we had to rely on culture tests, despite their low sensitivity, because culture positivity does not always confirm the presence of infection, owing to the undeniable fact that organisms may be either contaminants or colonizers, as described by Iqbal *et al.* [12]. It should be noted that colonization is distinguishable from infection: colonization means only the existence of microorganisms without any induction of disease or anomaly, whereas infection refers to the induction of disease or anomaly due to the presence of pathogenic microorganisms, which are the etiological agents of the caused disease [13, 53]. Further studies involving the latest

diagnostic techniques are required to overcome the gaps in terms of diagnosis, and standardized, multi-center studies should be conducted to overcome the lack of standardization.

## Conclusions

CPA had a significant association with PTB in the majority of the cases., The co-existence of CPA along with active PTB was also seen in some cases. In the later cases, treatment response rates were comparatively low. Mortality had a significant relationship with mean LOS, consolidation, hematological malignancies and stay at ICU, chronic renal failure, respiratory complications and DM. Increased mean LOS, hematological malignancies, presence of consolidation and stay at ICU were the independent factors associated with mortality. The unavailability of antigen detection techniques made diagnosis delayed and questionable. Further multi-center prospective studies involving antigen detection techniques are required for better understanding of the disease.

## Supporting information

**S1 Fig. Enrollment of patients having pulmonary aspergillosis.**
(TIF)

**S1 Dataset.**
(XLSX)

## Author Contributions

**Conceptualization:** Waqas Akram, Syed Azhar bin Syed Sulaiman, Amer Hayat Khan.

**Data curation:** Waqas Akram, Muhammad Bilal Ejaz, Amer Hayat Khan.

**Formal analysis:** Waqas Akram, Tauqeer Hussain Mallhi, Syed Azhar bin Syed Sulaiman.

**Investigation:** Waqas Akram, Muhammad Bilal Ejaz, Tauqeer Hussain Mallhi, Amer Hayat Khan.

**Methodology:** Waqas Akram, Muhammad Bilal Ejaz, Tauqeer Hussain Mallhi, Syed Azhar bin Syed Sulaiman, Amer Hayat Khan.

**Supervision:** Tauqeer Hussain Mallhi, Syed Azhar bin Syed Sulaiman, Amer Hayat Khan.

**Validation:** Tauqeer Hussain Mallhi, Amer Hayat Khan.

**Writing – original draft:** Waqas Akram, Tauqeer Hussain Mallhi.

**Writing – review & editing:** Waqas Akram, Tauqeer Hussain Mallhi, Syed Azhar bin Syed Sulaiman, Amer Hayat Khan.

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
