## [Decision Letter · Decision Letter 0]

1 Feb 2021

PONE-D-20-37556

Clinical Manifestations, Associated Risk Factors and Treatment Outcomes of Chronic Pulmonary Aspergillosis (CPA): An Experience from Tertiary Care Hospital in Lahore, Pakistan

PLOS ONE

Dear Dr. Akram,

Thank you for submitting your manuscript to PLOS ONE. After careful consideration, we feel that it has merit but does not fully meet PLOS ONE’s publication criteria as it currently stands. Therefore, we invite you to submit a revised version of the manuscript that addresses the points raised during the review process.

Please see comments made by the reviewers and provide point by point response in your revised manuscript. Manuscript will also benefit from expert native language editing prior to resubmission.

We look forward to receiving your revised manuscript.

Kind regards,

Muhammad Adrish

Academic Editor

PLOS ONE

Journal Requirements:

2.We suggest you thoroughly copyedit your manuscript for language usage, spelling, and grammar. If you do not know anyone who can help you do this, you may wish to consider employing a professional scientific editing service.  

3. Thank you for including your statement regarding patient consent from living patients. In your ethics statement in the manuscript and in the online submission form, please clarify whether any patients were unable to provide written informed consent regarding the use of the clinical records e.g. because they were deceased; if so, please state whether the IRB or ethics committee waived the requirement for informed consent from these patients.

4. In the ethics statement in the manuscript and in the online submission form, please provide additional information about the patient records/samples used in your retrospective study, including: a) whether all data were fully anonymized before you accessed them; b) the date range (month and year) during which patients' medical records/samples were accessed.

Reviewers' comments:

Reviewer's Responses to Questions

**Comments to the Author**

1. Is the manuscript technically sound, and do the data support the conclusions?

Reviewer #1: Yes

Reviewer #2: Partly

Reviewer #3: Partly

2. Has the statistical analysis been performed appropriately and rigorously? 

Reviewer #1: I Don't Know

Reviewer #2: Yes

Reviewer #3: N/A

3. Have the authors made all data underlying the findings in their manuscript fully available?

Reviewer #1: Yes

Reviewer #2: Yes

Reviewer #3: No

4. Is the manuscript presented in an intelligible fashion and written in standard English?

Reviewer #1: Yes

Reviewer #2: No

Reviewer #3: No

5. Review Comments to the Author

Reviewer #1: This is a fairly straightforward retrospective study of outcomes in pulmonary aspergillosis. Overall it provides helpful granular information regarding risk of mortality and complications for various types of pulmonary aspergillosis and as such is of value to the literature.

Major comments:

1. There are many grammatical and spelling errors and the document would benefit from an edit by a native english speaker.

2. Mortality is discussed but I don't see any discussion of the average duration of follow up for cases. Mortality is 100% if the follow up period is long enough.

3. I don't know if it adds anything to say that longer LOS and ICU admission predict mortality as hospitalization and ICU admission tend to proceed mortality and are outcomes themselves.

Reviewer #2: The manuscript by Akram et al descibes clinical characteristics of a cohort of patients with pulmonary aspergillosis. The strenght of the study is the large number of patients with postTB aspergilloma. However, the topic has been extensively reviewed in the last years and the manuscript does not add information to current knowledge. The english academic language needs some corrections.

Reviewer #3: The study investigated the clinical manifestations, associated risk factors and treatment outcomes of chronic pulmonary aspergillosis in a tertiary care hospital in Pakistan. It’s important. However, there are several issues in the study.

1.The dosage and course of steroid, and respiratory complications (Table 4) were not clearly described in the study.

2.Age distribution, treatment choice, and the impact on prognosis in patients with tuberculosis is not well analyzed.

3.Moreover, many factors such as underlying conditions and treatment medications may also impact the alarming mortality rate. More detailed analysis is needed.

6. PLOS authors have the option to publish the peer review history of their article (what does this mean?). If published, this will include your full peer review and any attached files.

Reviewer #1: No

Reviewer #2: No

Reviewer #3: **Yes: **Chao Wu

---

## [Author Response · Author response to Decision Letter 0]

12 Apr 2021

Reviewer 1: I have incorporated all of your suggestions into my revision. They were helpful. Thank you.

Reviewer 2: I have incorporated all of your suggestions into my revision. They were helpful. Thank you.

Reviewer 3: I have incorporated all of your suggestions into my revision. They were helpful. Thank you.

---

## [Decision Letter · Decision Letter 1]

6 May 2021

PONE-D-20-37556R1

Clinical Manifestations, Associated Risk Factors and Treatment Outcomes of Chronic Pulmonary Aspergillosis (CPA): An Experience from Tertiary Care Hospital in Lahore, Pakistan

PLOS ONE

Dear Dr. Akram,

Thank you for submitting your manuscript to PLOS ONE. After careful consideration, we feel that it has merit but does not fully meet PLOS ONE’s publication criteria as it currently stands. Therefore, we invite you to submit a revised version of the manuscript that addresses the points raised during the review process.

ACADEMIC EDITOR: Please review comments made by the reviewer and provide point by point response in your revised manuscript.. 

We look forward to receiving your revised manuscript.

Kind regards,

Muhammad Adrish, MD, MBA, FCCP, FCCM

Academic Editor

PLOS ONE

Journal Requirements:

Reviewers' comments:

Reviewer's Responses to Questions

**Comments to the Author**

1. If the authors have adequately addressed your comments raised in a previous round of review and you feel that this manuscript is now acceptable for publication, you may indicate that here to bypass the “Comments to the Author” section, enter your conflict of interest statement in the “Confidential to Editor” section, and submit your "Accept" recommendation.

Reviewer #2: All comments have been addressed

2. Is the manuscript technically sound, and do the data support the conclusions?

Reviewer #2: Partly

3. Has the statistical analysis been performed appropriately and rigorously? 

Reviewer #2: Yes

4. Have the authors made all data underlying the findings in their manuscript fully available?

Reviewer #2: Yes

5. Is the manuscript presented in an intelligible fashion and written in standard English?

Reviewer #2: No

6. Review Comments to the Author

Reviewer #2: The manuscript has been greatly improved. The description of statistical analysis needs further clarification regarding the multiple regression analysis and how parameters were selected. In the tables it should be clarified whether range, IQR or SD is presented.

The are still a number of spelling errors.

7. PLOS authors have the option to publish the peer review history of their article (what does this mean?). If published, this will include your full peer review and any attached files.

Reviewer #2: No

---

## [Author Response · Author response to Decision Letter 1]

17 Jun 2021

Respected Editor, 

 We have received revision for our submitted manuscript. We have hereby addressed all the concerns of the reviewer and all the suggestions made by the reviewer have been duly acknowledged. We are hopeful that the concerns of the reviewer will be satisfied by revised version of the manuscript. We have attached point-by-point response to the concerns of the reviewer and the revised version of manuscript has been highlighted in order to make the changes evident. Please let us know if there are any other suggestions or if any further changes are required. 

We are thankful to the editor and the reviewers for their time and efforts in putting their valuable suggestions and recommendations to make this manuscript more scientifically elegant and technically sound.

Journal Requirements:

Response: There is no retracted reference in our article. We did some changes to the reference list in the previous revision. Following are the new references that were added: 

Number in References References Added

25. Kosmidis C, Denning DW. The clinical spectrum of pulmonary aspergillosis. Thorax. 2015;70(3):270-7.

26. Ader F, Nseir S, Le Berre R, Leroy S, Tillie-Leblond I, Marquette C, et al. Invasive pulmonary aspergillosis in chronic obstructive pulmonary disease: an emerging fungal pathogen. Clinical Microbiology and Infection. 2005;11(6):427-9.

36. Uzunhan Y, Nunes H, Jeny F, Lacroix M, Brun S, Brillet P-Y, et al. Chronic pulmonary aspergillosis complicating sarcoidosis. European Respiratory Journal. 2017;49(6).

Apart from that, in this revision, a duplicated reference has been removed. The reference present at number 47 in the previous version, was also present at number 12. Below are the details of that particular reference: 

Number in previous version. Reference Removed

47. Iqbal N, Irfan M, Zubairi ABS, Jabeen K, Awan S, Khan JA. Clinical manifestations and outcomes of pulmonary aspergillosis: experience from Pakistan. BMJ open respiratory research. 2016;3(1). 

2. Please ensure that you refer to Table 4 in your text as, if accepted, production will need this reference to link the reader to the Table.

Response: The revised manuscript has followed the direction.

3. Please upload a copy of Supporting information Figure S1 which you refer to in your text on line 586.

Response: The revised manuscript has followed the direction.

Reviewer 2: Reviewer Comments

Reviewer: The manuscript has been greatly improved. The description of statistical analysis needs further clarification regarding the multiple regression analysis and how parameters were selected. In the tables it should be clarified whether range, IQR or SD is presented.Theere are still a number of spelling errors. 

Response: Respected reviewer, we are pleased to know that you have seen great improvements in our revised manuscript. Your concern towards our manuscript is really appreciated and we are thankful that you are taking pains to recommend further improvements. 

Concern 1: The description of statistical analysis needs further clarification regarding the multiple regression analysis and how parameters were selected.

Response 1: Respected reviewer, prior to performing the multivariate analysis, univariate analyses of the independent variables with the outcomes (that were stable/recovered, Deteriorated, and Mortality) were conducted. For the univariate analysis, all the parameters/variables that were clinically and biologically plausible, and all those indicated by previous literature, were selected. The variables reaching significance on univariate analyses, i.e., p ≤ 0.05, were selected for multinomial logistic regression analysis, in order to determine the independent factors associated with mortality. Furthermore, in regression output, the redundant parameters were excluded from the model. The results were presented in tables as p-value and odds ratio, along with 95% confidence intervals. For your convenience, we have highlighted the text in methodology, in which we have mentioned that how the variables were selected. 

Concern 2: In the tables it should be clarified whether range, IQR or SD is presented.

Response 2: Respected reviewer, the mentioned aspect has been clarified and highlighted in tables. 

Concern 3: Theere are still a number of spelling errors. 

Response 3: Respected reviewer, all the spelling errors have been corrected and the manuscript has been subjected to a review by a native speaker. We expect that the revised version will satisfy this concern. 

Respected reviewer,

We have tried our best to improve the manuscript and have made and highlighted the recommended changes in the manuscript. In this point-by-point response document, we did not list all the changes, but the changes are marked in the revised version. We are hopeful that this revision will meet with approval.

---

## [Decision Letter · Decision Letter 2]

15 Jul 2021

PONE-D-20-37556R2

Clinical Manifestations, Associated Risk Factors and Treatment Outcomes of Chronic Pulmonary Aspergillosis (CPA): An Experience from Tertiary Care Hospital in Lahore, Pakistan

PLOS ONE

Dear Dr. Khan,

Thank you for submitting your manuscript to PLOS ONE. After careful consideration, we feel that it has merit but does not fully meet PLOS ONE’s publication criteria as it currently stands. Therefore, we invite you to submit a revised version of the manuscript that addresses the points raised during the review process.

ACADEMIC EDITOR: Please review comments made by the reviewer and address them in your revised manuscript. Specifically language editing is required prior to acceptance of the manuscript.

We look forward to receiving your revised manuscript.

Kind regards,

Muhammad Adrish, MD, MBA, FCCP, FCCM

Academic Editor

PLOS ONE

Journal Requirements:

Reviewers' comments:

Reviewer's Responses to Questions

**Comments to the Author**

1. If the authors have adequately addressed your comments raised in a previous round of review and you feel that this manuscript is now acceptable for publication, you may indicate that here to bypass the “Comments to the Author” section, enter your conflict of interest statement in the “Confidential to Editor” section, and submit your "Accept" recommendation.

Reviewer #1: All comments have been addressed

Reviewer #2: All comments have been addressed

2. Is the manuscript technically sound, and do the data support the conclusions?

Reviewer #1: Yes

Reviewer #2: Yes

3. Has the statistical analysis been performed appropriately and rigorously? 

Reviewer #1: Yes

Reviewer #2: I Don't Know

4. Have the authors made all data underlying the findings in their manuscript fully available?

Reviewer #1: Yes

Reviewer #2: Yes

5. Is the manuscript presented in an intelligible fashion and written in standard English?

Reviewer #1: Yes

Reviewer #2: No

6. Review Comments to the Author

Reviewer #1: Explanation of the statistical analyses has greatly improved. The document is still somewhat difficult to read and it would benefit from a once over by a native english speaking editor, but I think the data and conclusions are sound.

Reviewer #2: The manuscript has been greatly improved. Comments have been addressed. However the academic language needs revision.

7. PLOS authors have the option to publish the peer review history of their article (what does this mean?). If published, this will include your full peer review and any attached files.

Reviewer #1: **Yes: **Brett Williams

Reviewer #2: No

---

## [Author Response · Author response to Decision Letter 2]

12 Aug 2021

Respected Editor, 

 We have received further revision for our submitted manuscript. The concerns of the reviewers have been duly acknowledged. We are hopeful that the concerns of the reviewers will be satisfied by this version of the manuscript. We have attached point-by-point response to the concerns of the reviewers and the revised version of manuscript has been highlighted in order to make the changes evident. 

We are greatly thankful to the editor and the reviewers for their time and efforts in putting their valuable suggestions and recommendations to make this manuscript scientifically elegant, technically sound and comprehendible. 

Reviewer Comments

Reviewer 1: Explanation of the statistical analyses has greatly improved. The document is still somewhat difficult to read and it would benefit from a once over by a native english speaking editor, but I think the data and conclusions are sound.

Response: Respected reviewer, we are pleased to know that you are satisfied with the explanation of statistical analyses. The manuscript has been subjected to review by a native English-speaking editor and Proofreading certificate is attached. We are hopeful that the revised version will satisfy this concern. 

Respected reviewer,

We have tried our best to improve the manuscript by subjecting it to an extensive review by a native English speaking editor. We are hopeful that this revision will meet with approval. 

Reviewer 2: The manuscript has been greatly improved. Comments have been addressed. However the academic language needs revision.

Response: Respected reviewer, we are pleased to know that you have seen great improvements in the manuscript. The manuscript has been subjected to review by a native English-speaking editor and Proofreading certificate is attached. We are hopeful that the revised version will satisfy this concern. 

Respected reviewer,

We have tried our best to improve the manuscript by subjecting it to an extensive review by a native English-speaking editor. We are hopeful that this revision will meet with approval.

---

## [Decision Letter · Decision Letter 3]

27 Oct 2021

Clinical Manifestations, Associated Risk Factors and Treatment Outcomes of Chronic Pulmonary Aspergillosis (CPA): Experiences from a Tertiary Care Hospital in Lahore, Pakistan

PONE-D-20-37556R3

Dear Dr. Khan,

We’re pleased to inform you that your manuscript has been judged scientifically suitable for publication and will be formally accepted for publication once it meets all outstanding technical requirements.

Kind regards,

Muhammad Adrish, MD, MBA, FCCP, FCCM

Academic Editor

PLOS ONE

Additional Editor Comments (optional):

All comments have been addressed

Reviewers' comments:

Reviewer's Responses to Questions

**Comments to the Author**

1. If the authors have adequately addressed your comments raised in a previous round of review and you feel that this manuscript is now acceptable for publication, you may indicate that here to bypass the “Comments to the Author” section, enter your conflict of interest statement in the “Confidential to Editor” section, and submit your "Accept" recommendation.

Reviewer #2: All comments have been addressed

2. Is the manuscript technically sound, and do the data support the conclusions?

Reviewer #2: Yes

3. Has the statistical analysis been performed appropriately and rigorously? 

Reviewer #2: Yes

4. Have the authors made all data underlying the findings in their manuscript fully available?

Reviewer #2: Yes

5. Is the manuscript presented in an intelligible fashion and written in standard English?

Reviewer #2: Yes

6. Review Comments to the Author

Reviewer #2: The manuscript has been greatly improved.

All relevant comments have been met.

The manuscript merits publication.

7. PLOS authors have the option to publish the peer review history of their article (what does this mean?). If published, this will include your full peer review and any attached files.

Reviewer #2: No

---

## [Editor Report · Acceptance letter]

3 Nov 2021

PONE-D-20-37556R3 

Clinical Manifestations, Associated Risk Factors and Treatment Outcomes of Chronic Pulmonary Aspergillosis (CPA): Experiences from a Tertiary Care Hospital in Lahore, Pakistan 

Dear Dr. Khan:

I'm pleased to inform you that your manuscript has been deemed suitable for publication in PLOS ONE. Congratulations! Your manuscript is now with our production department. 

Kind regards, 

on behalf of

Dr. Muhammad Adrish 

Academic Editor

PLOS ONE